# Microbiota, Microbial Metabolites, and Barrier Function in A Patient with Anorexia Nervosa after Fecal Microbiota Transplantation

**DOI:** 10.3390/microorganisms7090338

**Published:** 2019-09-10

**Authors:** Petra Prochazkova, Radka Roubalova, Jiri Dvorak, Helena Tlaskalova-Hogenova, Martina Cermakova, Petra Tomasova, Blanka Sediva, Marek Kuzma, Josef Bulant, Martin Bilej, Pavel Hrabak, Eva Meisnerova, Alena Lambertova, Hana Papezova

**Affiliations:** 1Laboratory of Cellular and Molecular Immunology, Institute of Microbiology, Academy of Sciences of the Czech Republic, v.v.i., Videnska 1083, 142 20 Prague 4, Czech Republic; r.roubalova@biomed.cas.cz (R.R.); dvorak@biomed.cas.cz (J.D.); tlaskalo@biomed.cas.cz (H.T.-H.); mbilej@biomed.cas.cz (M.B.); 2Laboratory of Molecular Structure Characterization, Institute of Microbiology, Academy of Sciences of the Czech Republic, v.v.i., Videnska 1083, 142 20 Prague 4, Czech Republic; martina.buganova@biomed.cas.cz (M.C.); petra.tomasova@biomed.cas.cz (P.T.); sediva@kma.zcu.cz (B.S.); kuzma@biomed.cas.cz (M.K.); 3Faculty of Chemical Technology, University of Chemistry and Technology Prague, Technicka 5, 166 28 Prague 6, Czech Republic; 44th Medical Department, First Faculty of Medicine, Charles University and General Faculty Hospital in Prague, U Nemocnice 2, 128 08 Prague 2, Czech Republic; 5Faculty of Applied Sciences, University of West Bohemia, Univerzitni 8, 306 14 Plzen, Czech Republic; 6Department of Psychiatry, First Faculty of Medicine, Charles University and General University Hospital in Prague, Ke Karlovu 11, 121 08 Prague 2, Czech Republic; josef.bulant@gmail.com (J.B.); Alena.Lambertova@vfn.cz (A.L.); Hana.Papezova@vfn.cz (H.P.); 7Department of Internal Medicine, First Faculty of Medicine, Charles University and General University Hospital in Prague, Ke Karlovu 11, 121 08 Prague 2, Czech Republic; Pavel.Hrabak@vfn.cz (P.H.); Eva.Meisnerova@vfn.cz (E.M.)

**Keywords:** microbiome, microbial metabolites, fecal microbiota transplantation (FMT), small intestinal bacterial overgrowth syndrome, short-chain fatty acids, Firmicutes/Bacteroides, *Akkermansia*

## Abstract

The change in the gut microbiome and microbial metabolites in a patient suffering from severe and enduring anorexia nervosa (AN) and diagnosed with small intestinal bacterial overgrowth syndrome (SIBO) was investigated. Microbial gut dysbiosis is associated with both AN and SIBO, and therefore gut microbiome changes by serial fecal microbiota transplantation (FMT) is a possible therapeutic modality. This study assessed the effects of FMT on gut barrier function, microbiota composition, and the levels of bacterial metabolic products. The patient treatment with FMT led to the improvement of gut barrier function, which was altered prior to FMT. Very low bacterial alpha diversity, a lack of beneficial bacteria, together with a great abundance of fungal species were observed in the patient stool sample before FMT. After FMT, both bacterial species richness and gut microbiome evenness increased in the patient, while the fungal alpha diversity decreased. The total short-chain fatty acids (SCFAs) levels (molecules presenting an important source of energy for epithelial gut cells) gradually increased after FMT. Contrarily, one of the most abundant intestinal neurotransmitters, serotonin, tended to decrease throughout the observation period. Overall, gut microbial dysbiosis improvement after FMT was considered. However, there were no signs of patient clinical improvement. The need for an in-depth analysis of the donor´s stool and correct selection pre-FMT is evident.

## 1. Introduction

Increasing research evidence is documenting the importance of the role of the gut microbiome in the regulation of behavior, mood, gastrointestinal symptomology, appetite, satiety, and nutrient metabolism. These are specific and non-specific core symptoms altered in anorexia nervosa (AN) patients. The studies have demonstrated significant changes in the microbial composition in individuals with AN in comparison with healthy or overweight individuals [1]. However, specific alterations vary between studies [2].

The most common bacterial species in human gut microbiota are members of the Firmicutes and Bacteroidetes phyla. The F/B ratio is increased in obese people compared to lean people, and tends to decrease with weight loss [3,4].

In connection with the altered microbiome in AN patients, particular bacterial species are mentioned, including the mucin-degrading bacteria *Akkermansia muciniphila* from the phylum Verrucomicrobia [5]. Its amount in the human intestinal tract is inversely correlated to several diseases and is considered as a new next-generation beneficial microbe [6]. Another species related to AN is an archaeon *Methanobrevibacter smithii*, which removes excess hydrogen from bacterial fermentation of polysaccharides. *M. smithii* was elevated in the microbiota of AN patients [7,8].

Gut dysbiosis is characterized mainly by the presence of pathobionts, decreased bacterial diversity, and an insufficient amount of beneficial bacterial species. It is often connected with impaired intestinal permeability and gut barrier dysfunction [9]. The presence of commensal intestinal bacteria plays an essential role in the development of both the native and adaptive immune system [10]. The altered composition of such commensal bacteria can lead to increased permeability of tight junctions to macromolecules, and results in the modification of the mucosal barrier function [9]. This so-called leaky gut is associated with the development of low-grade systemic inflammation, chronic diseases, sepsis, and also with some neurological and psychiatric diseases [11], and can be caused by starvation [12].

Gut microbiota produce short-chain fatty acids (SCFAs) during saccharide fermentation, represented mainly by acetate, propionate, butyrate, and valerate, utilized by colonocytes [13]. Relatively minor amounts of SCFAs are produced by the fermentation of protein-derived branched chain amino acids [14]. The increased SCFA levels were observed in obese and overweight people [15,16] while in AN patients, decreased levels of acetate and propionate were detected [17]. Microorganisms also produce various neuroactive substances, such as serotonin, dopamine, and norepinephrine [18]. Serotonin is one of the main neurotransmitters present in the brain. However, more than 90% of it is secreted by enterochromaffin cells in the gut, and thus interferes in the gut-brain axis bidirectional signalization [19].

Targeted microbiota modification represents an emerging therapeutic approach. Fecal microbiota transplantation (FMT), the administration of a fecal suspension from a healthy individual into the intestinal tract of another person, is a therapeutic procedure to manipulate the recipient´s microbiota. FMT is used mainly for the treatment of recurrent *Clostridium difficile* infection (rCDI) [20,21]. Except for rCDI, FMT was used to treat inflammatory bowel diseases (IBD) [22], diabetes mellitus [23], autistic spectrum disorders [24], obesity [16,25], and also to reduce symptoms of chronic intestinal pseudo-obstruction syndrome (CIPO) or small intestinal bacterial overgrowth syndrome (SIBO) [26,27]. SIBO is a digestive disorder with an increased count and/or abnormal spectrum of the small intestinal microbial population associated with different health conditions (e.g. immunodeficiency) common in AN patients [28].

Our patient case history with severe and enduring AN, refusing for years nutritional rehabilitation advice, and suffering from SIBO, demonstrates the impact of fecal transplantation on the microbiome and AN symptoms. The objectives of the present work were to investigate the change in the gut microbiome and microbial metabolites in an AN patient diagnosed with SIBO, and to evaluate the effect of FMT on the patient’s psychiatric conditions.

## 2. Materials and Methods

### 2.1. Patient

A 37 year old female patient, a psychiatric nurse, was recommended at 19 years to the outpatient facility specializing in eating disorders (ED). Her anorexia nervosa symptoms appeared at 13 years, and were diagnosed at her first hospitalization after a collapse with seizures (glycemia 2.5) when she achieved her least weight (37 kg/173 cm, BMI 12.36). She has a history of long-term repeated vomiting, diets, meat rejection, vegetarianism, abuse of laxatives, drinking of some bleach, and excessive exercise. Always skeptical about the treatment, she expressed the feeling she can live with the disease because without it she would lose her self-image. During the years of illness, she experienced various psychotherapy and pharmacotherapy (venlafaxin, hypnogen, seropram, prothazin, rivotril, prothiaden, fevarin), and had been hospitalized as an inpatient at the day care of a specialized ED center. When she reached 60 kg/173 cm (BMI 20.05), she felt "fat as a pig", and focused only on weight loss and gradually increasing somatic difficulties. Since 2016, the patient was treated for dyspepsia and malnutrition. Dysmicrobia with dyspeptic syndrome, mycotic upper esophagitis, hooked stomach, stomach evacuation disorder, and duodenogastric reflux were indicated. At that time, galactosemia and SIBO were diagnosed by an immunohistochemical examination of jejujan mucosa samples and jejunal fluid aspirate cultivation. SIBO was manifested by a wide range of symptoms: Bloating, diarrhea, constipation, abdominal pain, and abdominal cramps. Despite various antibiotic treatment regimes, and a low saccharide diet, only mild and temporary symptom alleviation was achieved. Due to the persistent symptomatology which limited the patient in daily activities, and considering the negative impact of the SIBO symptomatology, as well as the negative impact of the low saccharide diet on AN, the FMT was applied (approved by the Ethical Committee of the University Hospital for severe gastrointestinal disorders). There were no known contraindications for this procedure. During the study, the authors monitored her psychiatric symptoms, family dynamics, and specific symptoms with eating disorder examination questionnaire (EDEQ). 

### 2.2. Donor Selection

The donor was a 67 year-old 1st-degree female relative of the patient. The donor was examined according to current Czech guidelines for fecal bacteriotherapy [29]. Briefly, the donor had no history of gastrointestinal diseases or other severe conditions (e.g., chronic diarrhea, constipation, colorectal polyps, cancer, immunocompromised states, obesity, diabetes, inflammatory bowel disease, irritable bowel syndrome, allergy, or gastrointestinal surgery), and did not use antibiotics or any other medications or food supplements in the past three months. The donor underwent laboratory evaluation, including a complete blood count, C-reactive protein, erythrocyte sedimentation rate (FW), and biochemical tests including the detection of infectious communicable disease agents (blood tests for HIV, hepatitis A, B, and C, syphilis, presence of Epstein-Barr virus, and cytomegalovirus). In the feces, natural intestinal flora was confirmed. The donor stool was tested for adenovirus, norovirus, enterovirus infection, and *C. difficile*, ova, and parasites. Both the donor and the recipient subscribed informed consents.

### 2.3. Stool Preparation

Firstly, 250 grams of freshly acquired stool (within 4 hours) was homogenized in a home blender with approx. 600 mL of 10% glycerol solution and filtered through sterile gauze to remove solid matter. The homogenized stool was divided into three equal doses. One was immediately applied to the patient, while the other two were stored at –80°C and thawed before further application.

### 2.4. Stool Transplant

Before application of the stool transplant, the recipient had nothing to eat or drink for 6 hours. Preceding the rectal application, a saline enema was administered to the patient. The first sample was applied via the working channel of gastro duodenoscopy deep to the descending duodenum. The second stool administration was executed similarly 4 days later. After two weeks, the last stool dose was applied via a nasogastric tube to the sigmoid colon. No adverse effect of FMT during the procedures was registered. The patient complained about a change of stool frequency from constipation to diarrhea, which disappeared spontaneously, similarly to other cases [30]. The patient’s stool was collected prior to FMT, at 2 weeks and 1, 2, 3, 5, 6, and 12 months after the first FMT. Further, the donor stool was used in a microbiota analysis. Additionally, the patient serum was collected before FMT and over the course of 6 months.

### 2.5. I-FABP Levels

The serum concentrations of intestinal fatty acid-binding protein (I-FABP) were analyzed in duplicates using a highly specific, commercially available enzyme-linked immunosorbent assay (ELISA) that selectively detects human I-FABP (HyCult Biotechnology, Uden, The Netherlands) The assay was performed according to the manufacturer’s recommendations, with serum samples diluted 1:5 in the provided calibrator diluent. The concentration was determined via a standard curve from the absorbance read at 450 nm. The serum was also used for the analysis of common biochemical markers including CRP, triacylglycerides, cholinesterase, albumin, thyroid-stimulating hormone, and free thyroxine (fT4).

### 2.6. Intestinal Microbiota Quantification by qPCR

The abundances of specific intestinal bacterial groups were assessed by qPCR (CFX96 Touch, Bio-Rad) using universal (all bacteria) or species-specific primer sets (*Akkermansia muciniphila, Methanobrevibacter smithii*, Appendix A). The amplification conditions and plasmid preparation are described (Appendix A). The plasmids containing a corresponding 16S rRNA gene insert were used as the standard to validate the absolute copy number of each sample in a 10-fold dilution series. Analyses of changes in total bacterial number and *A. muciniphila* and *M. smithii* colonizations were performed using a one-way analysis of variance with Dunnett post-test, with a *P* value of <0.05 being considered significant.

### 2.7. High Throughput Sequencing of Microbiota Composition

The total extracted genomic DNA (gDNA) from stool samples was used for high throughput sequencing (HTS, Miseq platform, Illumina, San Diego, CA, USA) of the bacterial V4 region of the 16S rRNA gene, and fungal ITS region. The library preparation and sequence treatments are described in the Appendix A. The dataset obtained in this study was deposited in the NCBI Sequence Read Archive (raw demultiplexed sequencing data with sample annotations, PRJNA542274). The pipeline SEED v2.1 [31] (Institute of Microbiology, Prague, Czech republic) was used for α-diversity calculation (Shannon-Wiener index, species richness, Chao 1 index) using 35,000 (bacterial) or 25,000 (fungal) randomly selected sequences from each sample. The non-metric multidimensional scaling (NMDS) ordination method was used to analyze the bacterial and fungal community structure of the single samples using quantitative Bray-Curtis dissimilarity matrices. The analyses were performed using PAST v.3.15 [32] (University of Oslo, Oslo, Norway).

### 2.8. Metabolomic Analysis

Nuclear magnetic resonance (NMR) and mass spectrometry (MS) were used for the targeted metabolomic analysis of stool samples. The NMR analysis was used to measure SCFA levels, and an MS analysis was employed for the assessment of metabolites of the tryptophan pathway in the intestine. The preparation of samples for the NMR and MS analysis, as well as the data processing and evaluation, are described in the Appendix A.

## 3. Results

### 3.1. Patient’s Health Condition

During the whole study, the patient was very collaborative, expecting mainly the improvement of increasing gastrointestinal problems. Over the course of 12 months, she maintained at 52–54 kg (BMI 17.4–18.4), vomited at different frequencies, returned to restrictions, and planned to discard dairy products completely to reduce the problems with bloating and diarrhea. She did not change her diet in accordance with her nutritionist, nor did she change her dietary habits while visiting abroad (only egg whites and legumes, mixed fruit). She started to vomit. She rationalizes non-compliance with her nutritionist by denying gradually their association with purgative symptoms of her AN. One year after the FMT, the patient still suffers from insufficient stool passage and digestion complaints. However, she consumes a very-low saccharide diet, approaching nearly no food.

### 3.2. Changes in Intestinal Barrier Function

In order to analyze the patient’s intestinal barrier function, serum I-FABP levels were assessed (Figure 1), and were significantly increased in the patient before (863.05 versus 288.11 pg/mL; *p* = 0.017) and 14 days after the first FMT (581.58 versus 288.11 pg/mL; *p* = 0.049), compared to the control samples. There was a steady decrease in I-FABP levels within 6 months post-FMT with non-detectable values in the final sample. The sera of sixty healthy volunteers (women aged 18–35), recruited for a concurrent project concerning microbiome changes in AN patients, were used as the control samples. Apart from the I-FABP serum levels, all common measured biochemical parameters (CRP, triacylglycerides, cholinesterase, albumin, TSH, and fT4) were in the physiological range during the whole 6 month period.

### 3.3. Quantification of Intestinal Microbiota by qPCR

To measure total bacterial load and the absolute amount of *A. muciniphila* and *M. smithii* in the stool samples, qPCR was employed. The 16 rRNA assessment of the bacteria present did not detect significant changes throughout the measurement period. Thus, the FMT procedures did not substantially influence the gut bacterial load (Figure 2A). Further, the amount of beneficial *A. muciniphila* did not change considerably during the 6 months of measurement, and at 12 months post-FMT, significant increase in *A. muciniphila* was detected (Figure 2B). Interestingly, any *A. muciniphila* DNA in the donor sample were not detected. Similarly, the methanogenic archaeon *M. smithii* 16S rRNA copies were not altered except the last sample 12 months post-FMT, which was significantly increased. Furthermore, a similar level of *M. smithii* was found in the donor sample (Figure 2C).

### 3.4. Microbial Community Taxonomic Composition

The microbiome analysis by HTS resulted in 1,153 bacterial OTUs classified into 8 phyla and 156 genera, and in 754 fungal OTUs classified into 5 phyla and 270 genera (Figure 3). During the whole tested period, the patient gut bacterial community was dominated by Bacteroidetes (37–72% from all microbial communities), and by Firmicutes (25–54% from all microbial communities), followed by Actinobacteria, Proteobacteria, and to a lesser extent by Verrucomicrobia, Cyanobacteria, Tenericutes, and Lentisphaerae (Figure 3A). The FMT caused a substantial bacterial shift. At the phylum level, the main shift was seen in Bacteroidetes, where the abundance after 1 year post-FMT decreased from 72 to 45%. On the contrary, Firmicutes abundance increased from 25 to 47%. Similarly, Actinobacteria (from 1.3 to 5.6%) and Proteobacteria (from 1 to 6.1%) were elevated significantly. The F/B ratio shifted from extremely low values from very first samples (pre-FMT and early after FMT, 0.36–0.57), to values of approximately 1, which corresponds to the ratio common for healthy persons (Table 1). 

At the bacterial species level, there were two prominently abundant taxa – *Prevotella copri* and *Bacteroides* spp. *P. copri* which represented 56% of the total bacterial abundance in the patient stool sample before FMT. Its amount decreased to 6% after 6 months, and again tended to increase within the next 6 months post-FMT (Figure 3C). Interestingly, this bacteria was not detected in the donor. 

The abundance of the second prominent bacterial group (*Bacteroides* spp.) varied and the greatest amount was detected one-month post-FMT (from 11.6% to 21%), and at 12 months, its abundance slightly decreased to 15.8% (Figure 3C). The FMT led to the increased relative abundance of selected genera from order Clostridiales: *Roseburia* spp., *Ruminococcus* spp., *Blautia* spp. *Faecalibacterium prausnitzii, Clostridium* spp., and *Anaerostipes* spp., although the greatest increase was seen at different time points for each genus (Figure 3C,D). Particularly *Roseburia* spp., the dominant butyrate-producing Firmicute was present in very low levels pre-FMT. At 5 months post-FMT, there was a significant increase in its abundance, which declined after 12 months (Figure 3C). Contrary to the donor sample (3.2%), another common beneficial genus *Lactobacillus* spp. was present in very low levels during the experimental period, suggesting an insufficient abundance of commensal Lactobacillus species in the patient’s intestine (0–0.7% during the year). This can rehabilitate homeostasis in intestinal disorders or regulate the composition of the gut microbiota and correct abnormal responses of the mucosal immune system to chronic gut inflammation [33].

The comparison of genera in the stool of the patient and donor showed that FMT increased the abundance of some Clostridiales species (*Anaerostipes* spp., *Clostridium* spp., *F. prausnitzii*), but there was no significant shift in the overall composition of selected clostridium genera (Figure 3D).

The fungal community was constituted mainly of genera belonging to Ascomycota (from 77% to 99.7%), followed by Basidiomycota (from 13.3% to 0.2%), Mucoromycota (4.3% to 0.02%), and to a lesser extent, by Cryptomycota and Chytridiomycota (Figure 3B). In the patient sample pre-FMT, 5% of OTUs were not identified (Figure 3B). The diet strongly influences the fungal community composition, and therefore the irregular course of the relative abundances of particular fungal species have been seen (Appendix A). Interestingly, yeasts *Candida* and *Saccharomyces* spp. evinced similar abundance patterns with a strong decrease two months post-FMT. *Penicillium* spp. had a reverse pattern, and was greatly abundant at this 2-month interval. These results suggest some type of fungal competition. Further, the patient’s stool before FMT contained a relatively great amount of *Trichosporon* spp., a potential pathogen (Appendix A).

### 3.5. Alpha and Beta Diversity

Both bacterial and fungal rarefied sequences were used for the alpha-diversity calculation. A steep increase in bacterial diversity was observed as a result of FMT depicted by various alpha-diversity estimators (Figure 4A). The Shanon-Wiener index, accounting for both the abundance and evenness of the species present, as well as species richness, increased continuously until 6 months post-FMT. Similarly, the Chao1 index, the diversity estimator from the abundance data, was elevated throughout the whole year. There was also a small decrease in the richness parameters immediately after the FMT, most likely as a result of saline enema administration followed by bacterial reduction (Figure 4A).

The patient’s stool prior to the FMT contained a great number of fungal species (Figure 4B). This is in concordance with the patient’s SIBO diagnosis, caused mainly by fungal overgrowth. Contrarily to bacteria, a sudden decrease in fungal diversity was observed as a result of FMT. All three estimators were reduced substantially, indicating the improvement of the altered microbiome.

The NMDS ordination of the bacterial and fungal beta diversity in the donor and patient stool samples based on the dissimilarity measurement between the samples by comparing overlapping OTUs and their relative abundances is shown (Figure 5). Post-FMT, a shift in the microbial community composition of both the bacteria and fungi was evident. A more noticeable shift towards the composition of the donor stool was detected for fungi. This is probably a consequence of the easier manipulation of the fungal community. The patient’s bacterial community resembled more the situation pre-FMT than the donor´s community over the course of observation. 

### 3.6. Metabolomic Analysis of Microbial Metabolites

The levels of total SCFAs (i.e. acetate, butyrate, and propionate) were increased 1-year post-FMT compared to the initial values (Figure 6A). However, the particular content of SCFAs fluctuated over the course of the 1-year period post-FMT. The levels of individual SCFAs are depicted in the Appendix A. The relative SCFA concentrations in the stool were substantially reduced after the first FMT, probably mirroring the situation before the attachment of the transplanted microbes. The second FMT was followed by a considerable increase in all assessed SCFAs. Interestingly, the SCFA levels in the donor were less than the levels in the patient pre-FMT. 

The correlation between the SCFA levels and the abundance of selected bacterial species from Clostridiales order may suggest a relevant role of these bacteria in the production of SCFA. Nevertheless, other bacteria species also contribute to the pool of SCFA production (Figure 7).

Unexpectedly, the serotonin level measured in the stool samples had a downward trend (Figure 6B). Furthermore, the serotonin level in the donor stool sample was much less compared to the control samples from healthy women from a concurrent project (data not shown). The levels of tryptophan, 5-hydroxytryptophan, and kynurenine are depicted in the Appendix A.

## 4. Discussion

The intestinal epithelium allows the limited passage of microorganisms and their metabolites from the luminal space to the systemic compartment. The gut microbiota alteration may disrupt the intestinal barrier integrity, which can lead to increased intestinal permeability and gut barrier dysfunction. These pathological changes were detected in patients with IBD, various autoimmune diseases, liver cirrhosis, severe acute pancreatitis, and other metabolic diseases [34].

An elevated I-FABP serum concentration was detected in the patient before and 14 days post-FMT procedure, as compared to the healthy controls. This suggests the patient potentially suffered from some degree of intestinal damage. After 1-month post-FMT, the I-FABP values decreased to the values in the healthy controls and then further declined. Clearly the FMT had a positive effect on the altered barrier function of the AN patient. FMT can reduce gut dysfunction symptoms by increasing SCFA production, especially butyrate, which maintains the epithelial barrier integrity [35]. However, the SCFA levels, including butyrate, fluctuated over the 12 months post-FMT, reflecting the varying abundance of particular butyrate-producing bacteria. 

The total bacterial load in all patient samples, as well as in the donor sample, were similar and this was not affected by FMT (Figure 2A). The SCFA-producer *A. muciniphila* utilizes mucins as an energy source, which enables them to persist and colonize the mucus layer surface. Commonly, it accounts for 1 to 4% of intestinal bacteria in healthy adults [36], and its greater levels are considered as having a beneficial impact on human health. This study detected only a very small amount of *A. muciniphila* in the patient stool samples with a slight increase 12 months post-FMT (Figure 2B). Importantly, *A. muciniphila* in the donor’s stool was under the PCR detection limit, suggesting the poor microbial quality of the sample. The HTS data showed the *A. muciniphila* abundance fluctuated deeply under 1% (from 0.001 to 0.48% at 12 months post-FMT), most likely due to the patient’s nutrient- and energy-deprived gut environment.

The archaeon *M. smithii* was elevated in AN patients [37]. However, only a slight increased abundance 12 months post-FMT was detected, which is probably related to overall microbial transition after a longer period.

The human gut microbiota is dominated by representatives of two major phyla, Firmicutes and Bacteroidetes. In humans, an increased F/B ratio was found in obese individuals compared to normal-weight and lean adults [4,38], with the lowest F/B ratio (0.7) reported in persons with a BMI <18.5. Our AN patient with BMI 18.1 had a F/B ratio of 0.36 pre-FMT (Table 1), and then it slowly increased toward the F/B ratio considered as beneficial in healthy people. The low ratio observed pre-FMT was caused by the great abundance of Bacteroidetes, which accounts for almost 72%. Out of this phylum, 55.6% was represented by *Prevotella copri* (Figure 3C). Some *Prevotella* strains promote low-grade systemic inflammation and may be clinically important pathobionts [39].

The FMT resulted in an increase of specific genera within the order Clostridiales, particularly *Roseburia* spp, *Ruminococcus* spp., *Blautia* spp., *Faecalibacterium prausnitzii*, *Clostridium* spp., *Anaerostipes* spp., *Eubacterium* spp., although some of them only temporarily (Figure 3C). A similar increase in relative species abundance as a consequence of FMT, was shown previously [40]. Dominant butyrate-producing bacteria *Roseburia* spp., *F. prausnitzii,* and *Eubacterium rectale*, account for 7–24% of the total bacteria in the healthy human gut [41]. These species were increased mainly within 14 days and 2 months post-FMT, which is in agreement with a greater amount of SCFAs at 2 months post-FMT (Figure 6A). Further, *Roseburia* spp. contributes to the control of gut inflammatory processes, and its abundance was reduced in patients with inflammatory diseases [42], as well as in AN patients [8].

The fungal community accounts for approximately 0.1% of the human gut microbiome. The mycobiome analysis is much less conclusive than the bacterial microbiome, since its composition is strongly affected by diet. Mycobiome dysbiosis is often connected with different diseases, e.g. IBD, when the inflamed mucosa exerts a significantly increased fungal diversity and richness [43].

By the metabolomic analysis, total SCFA levels increased one-year post-FMT compared to the pre-FMT values (Figure 6A). The fluctuations in SCFA levels could reflect the varying abundance of individual SCFA producing microbes post-FMT (Figure 3C,D). The SCFAs are typically produced by several microbial species (*Roseburia* spp., *Clostridium* spp., *F. prausnitzii, Prevotella* spp., etc.). An increase in acetate and butyrate was also observed in a similar case study on anorectic patients after FMT [44]. The differences between the donor and patient pre-FMT may be associated with the abundance of the SCFA-producer *A. muciniphila* [44] as well *Prevotella copri*, which were not detected in the donor stool sample, but present in the patient. 

Serotonin secretion and the role of gut microbiota during its synthesis in the intestine is well described [19,45]. To the best of the patients’ knowledge, there is no available data to compare the effect of FMT on serotonin levels in AN patients. A downward trend was observed in serotonin levels post-FMT in the AN patient’s stool samples (Figure 6B). Similar FMT effects were described in rat intestines on a high-fat diet [46]. Additionally, the decreased expression of tryptophan hydroxylase 1, known as the rate-limiting enzyme in serotonin synthesis, was observed. Further, the increased expression of a serotonin transporter, which transports serotonin from the gastrointestinal tract to distribute it to peripheral tissues, after transplantation of feces from healthy volunteers into mice was reported [47]. It could be speculated that FMT affected the expression of the tryptophan hydroxylase 1 protein or serotonin transporter in the AN patient. The bacteria affect the secretion of serotonin through influencing enterochromaffin cells. This effect was observed mainly in spore-forming bacteria, such as *Clostridium*, in the gut microbiome [48]. However, no correlation was observed between the bacteria abundance and the level of serotonin in this study.

A diverse and stable gut microbiome correlates with a healthy intestine. Thus, FMT success is highly dependent on the microbial diversity and composition of the stool donor [49]. The existence of so-called FMT super-donors, whose stool leads to the most successful FMT outcomes, was discussed [50]. Usually, the donors are clinically screened for pathogens or disease occurrence, but not for microbial diversity, which is a crucial component in FMT success. Thus, the microbial analysis of donor stool could shift the success rate in the treatment of various diseases by FMT. From our case study, FMT can be considered as a promising therapeutic option for gut dysbiosis caused by SIBO, resulting from severe and enduring AN.

Pre-FMT, the patient evinced considerable signs of dysbiosis, comprising of very low bacterial and excessively great fungal diversity (fungal overgrowth) (Figure 4). As a result of FMT, bacterial diversity increased substantially and persisted for 1 year. Our case study of severe chronic AN does not allow the differentiation of whether microbiome changes were caused by long-term laxative abuse, vomiting, or by another specific eating pathology. The attitude, mood, or eating pattern of purgative AN was unchanged, despite significant improvement in the microbiome post-FMT. Her psychometric properties were evaluated with several questionnaires. The clinical observations were supported by EDE-Q, BDI II, BAI results from the initial and all the follow-up periods. Her clinical status (gastrointestinal pain and complaints despite all kinds of diets recommended by nutritionist and self-implicated restrictions and other specific and non-specific psychiatric symptoms) remained unchanged during all the observation period at out-patient visits.

The question remains, in short, a duration of the illness with an impact on gastrointestinal symptoms, could FMT be beneficial or could the microbiome analysis be used to predict disease chronification? Further studies will inform on the characteristics of nutritional rehabilitation to provide personalized AN realization protocols [51,52].

## Figures and Tables

**Figure 1 microorganisms-07-00338-f001:**
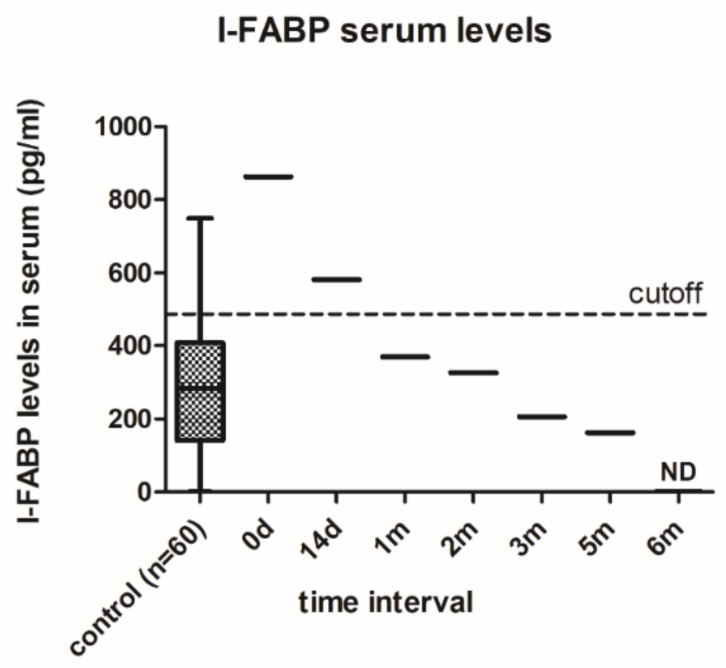
Serum I-FABP levels in the anorexia nervosa (AN) patient before (0 d) and after fecal microbiota transplantation FMT (14 d–6 m) compared with healthy subjects. Control data are presented as box-whiskers with median ± range (*n* = 60). The cutoff was determined based on FDR values <0.05.

**Figure 2 microorganisms-07-00338-f002:**
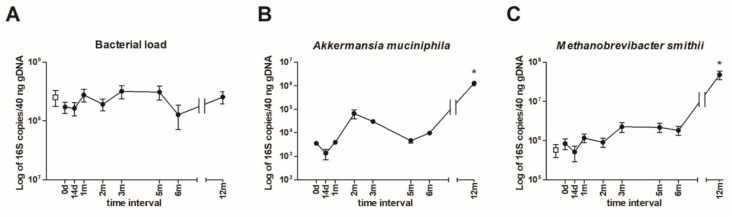
The qPCR quantification of the total amount of *Akkermansia muciniphila* bacteria, and archaeon *Methanobrevibacter smithii* in the donor and patient stool samples before (0 d) and after fecal microbiota transplantation (FMT) (14 d–12 m). (**A**) The total amount of bacterial 16S rRNA copies (**B**) *A. muciniphila* abundance (**C**) *M. smithii* abundance. The patient´s values are represented by black circles, and the donor values are represented by the empty squares (under the detection limit in **B**). The data are presented as the mean ± SEM. **p* < 0.05 represents a significant difference from the patient’s stool sample pre-FMT. The statistical significance was determined by one-way analysis of variance with Dunnett post-test.

**Figure 3 microorganisms-07-00338-f003:**
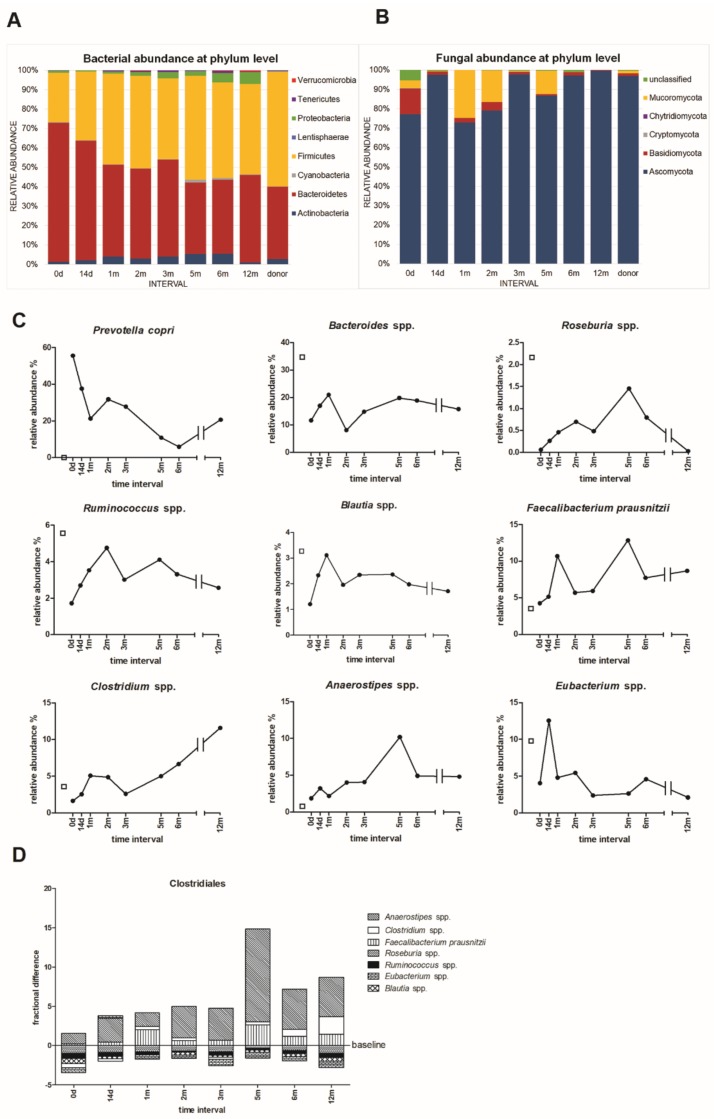
Taxonomy of the microbial communities of the donor and patient stool samples before (0 d) and after FMT (14 d–12 m). (**A**) The proportions of bacterial and (**B**) fungal phyla. (**C**) The relative abundance of assorted bacterial species present in stool samples. The patient´s values are represented by black circles, donor values are represented by the empty squares. (**D**) The relative fractional differences in the proportions of selected bacterial genera from order Clostridiales in the stool samples of the patient over the course of 12 months. The data are correlated using the donor sample values (baseline). The values above the baseline show a proportional increase in the abundance of particular bacterial genera compared to the donor’s sample. The bars under the baseline show a proportional decrease compared to the donor’s values.

**Figure 4 microorganisms-07-00338-f004:**
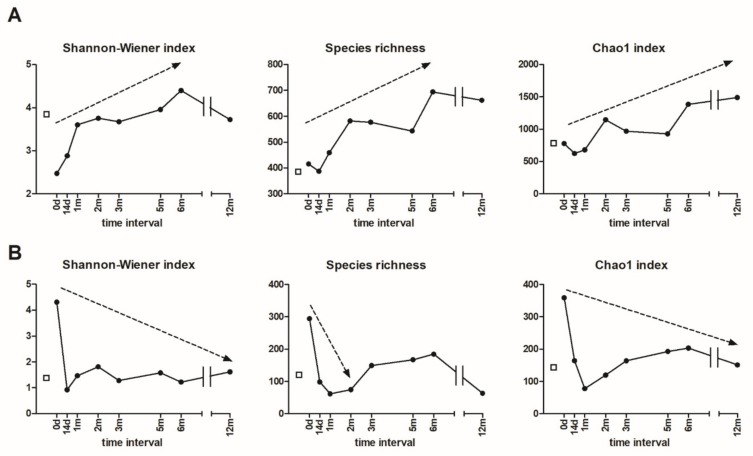
Alpha diversity of both (**A**) bacterial and (**B**) fungal sequences from the donor and patient stool samples before (0 d) and after FMT over the course of 12 months (14 d–12 m). The patient´s values are represented by black circles, the donor values are represented by the empty squares.

**Figure 5 microorganisms-07-00338-f005:**
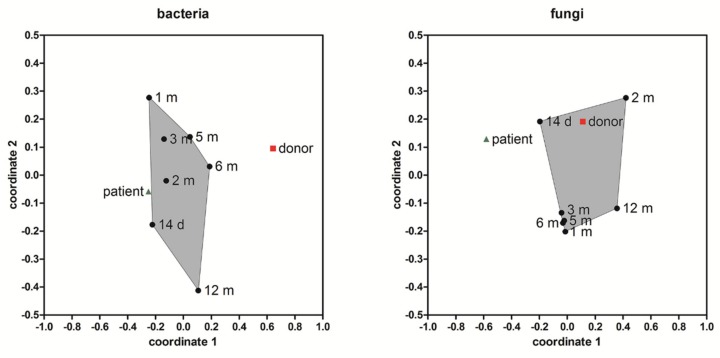
Non-metric multidimensional scaling (NMDS) ordination method was applied to visualize the similarity level of the bacterial and fungal community compositions from stool samples of the donor and patient. Convex hulls gather samples from the patient from 14 days to 12 months post-FMT (black circles), (green triangle = patient before transplantation, red square = donor). The stress value = 0.1054 (bacteria); 0.1338 (fungi).

**Figure 6 microorganisms-07-00338-f006:**
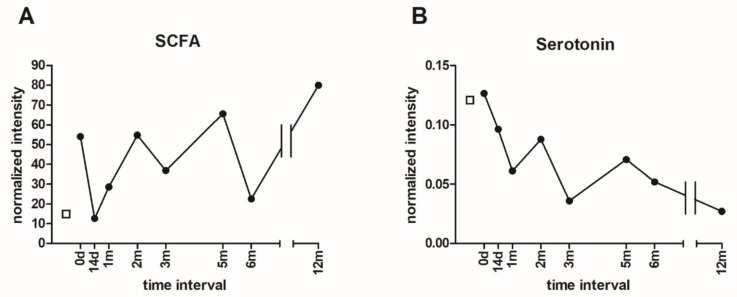
The short-chain fatty acids (SCFA) and serotonin levels in donor and patient stool samples before (0 d) and after FMT (14 d–12 m). (**A**) The total SCFA levels were determined by NMR (**B**) Serotonin levels were assessed by MS. The patient´s values are represented by black circles, donor values are represented by the empty squares.

**Figure 7 microorganisms-07-00338-f007:**
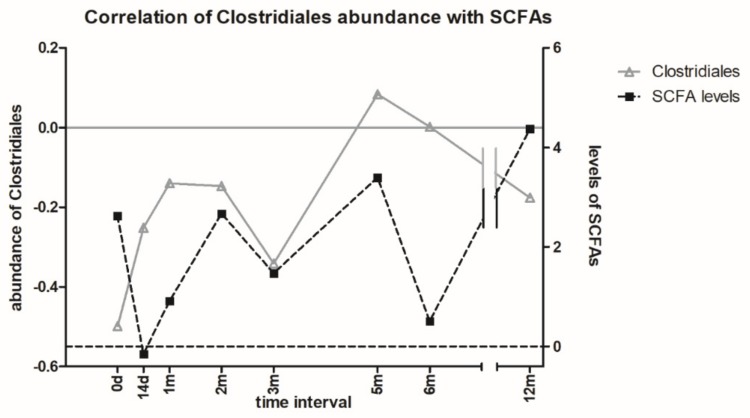
The correlation of Clostridiales abundance with SCFA levels in the patient stool samples over 12 months. The data are correlated using the donor sample values (baselines). After the FMT, SCFA levels correlated with Clostridiales abundance.

**Table 1 microorganisms-07-00338-t001:** Firmicutes/Bacteroidetes ratio in the stool donor and patient samples in time.

Donor	1.59						
Patient							
0 d	14 d	1 m	2 m	3 m	5 m	6 m	12 m
0.36	0.57	0.99	1.03	0.83	1.46	1.30	1.03

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
