# Peer review of "Microbiota, Microbial Metabolites, and Barrier Function in A Patient with Anorexia Nervosa after Fecal Microbiota Transplantation"

_microorganisms, 2019, doi:10.3390/microorganisms7090338_

Round 1

Reviewer 1 Report

The authors present a manuscript on a case report for FMT on an anorexia nervosa patient. While this is highly interesting, the study raises some questions.

Why did the authors use qPCR for only two species? Why not using only NGS? Metabolome studies on SCFA are not real metabolome studies, but rather studies in SCFA. Metabolome is much more. Why did they only look for serotonin? Why didn’t the authors connect the serotonin to specific microbes and their respective numbers? Why did the authors use i-FABP and not classical integrity markers like zonulin, alpha 1-antitrypsin or even calprotectin? The authors write several times within the manuscript, that Akkermansia muciniphila produces butyric acid. This is not true. Please see: https://doi.org/10.1099/ijs.0.02873-0 The often mentioned F/B ratio is scientifically not sound: Finucane et al. 2014 https://doi.org/10.1371/journal.pone.0084689 For many statements within the text, the respective literature is missing (eg. Page 2, l 47; p 2, l 58; p 2, l 59; p2, l 65) In figure 2 the points show a standard deviation. However, the authors state in the Supplement section, that they performed a double measurement only. Hence, a standard deviation is not possible, for this purpose one needs at least three measurements. Figure 2, B: Within the text the authors state, that there was no significant difference (p 5, l 208: slight increase). However, in the figure there is an asterix, an indicator for significance? Furthermore, an increase by 3 log shall be significant. Same applies for figure 2, C Part 3.4 on page 6, line 220 ff: The phylum is Bacteroidetes not Bacteroides! The % in brackets represents the variations during the measurement of 6 respectively 12 months. Please indicate so. Page 2, line 227: The shift was seen in Bacteroidetes or Bacteroides? Page 2, line 247: “insufficient probiotic bacteria” what are these? By definition a probiotic organism is taken by oral route. It is not a taxonomic group! Page 9, lines 301 ff: The authors did not measure the metabolome but merely SCFA concentrations. Page 9, Figure 6: What is a normalized intensity. SCFAs are measured in mM Page 10, Figure 7: Why do the authors compare Clostridiales with the SCFA concentration? SCFAs such as Acetate or Propionate are produced by members of all phyla, however butyrate is only produced by members of the Firmicutes. But NOT by members of the Verrucomicrobia as stated in the text!

In conclusion: the authors mix too many observations (F/B ratio, SCFA, qPCR, NGS, fungi etc.). The general “red story line” is missing.

Author Response

Response to Reviewer 1 Comments

We would like to thank to both reviewers for all comments, questions and for pointing out all discrepancies. We made efforts to answer all questions and correct manuscript according to reviewer’s suggestions.

1) Why did the authors use qPCR for only two species? Why not using only NGS?

Response 1: Data from NGS are reliable and definitely, they represent the main source of our results. However, the NGS analysis produces data about relative abundance of bacterial species. By qPCR, we are able to count the absolute amount of 16S rDNA of particular bacterial species. Both followed species (A. muciniphila and M. smithii) were earlier described to be somehow connected to microbiome of anorexia nervosa patients. Since A.m. is considered as a possible “probiotic of next generation” and its decrease is connected to couple of inflammatory diseases as well as to obesity, we believed that its assessment by another method could provide additional results to NGS (similarly to M. smithii, which was found to be increased in AN patients).

2) Metabolome studies on SCFA are not real metabolome studies, but rather studies in SCFA. Metabolome is much more. Why did they only look for serotonin? Why didn’t the authors connect the serotonin to specific microbes and their respective numbers?

Response 2a: The reviewer is undoubtedly right. Our study is not a study of the metabolome in the real sense of the word. However, it uses the common metabolomic platforms (NMR, MS) to study a limited set of metabolites, i.e. SCFA, serotonin and the metabolites of tryptophan pathway. To avoid any confusion, we exchanged the word “metabolome” for “microbial metabolites” in the article, which more precisely expresses our metabolomic focus.

b) Why did they only look for serotonin?

Response 2b: As we studied anorexia and its metabolic signature, the main focus was on the neurotransmitter serotonin. Except for serotonin, we also measured the levels of 5-hydroxytryptophan, kynurenine, and tryptophan as these all are the key metabolites of tryptophan metabolism, which includes serotonin production. Because the trends in tryptophan and kynurenine levels are mutually similar but different from the serotonin and 5-hydroxytryptophan, we hypothesized that the concentration of tryptophan is not limited factor for serotonin production in our case. Thus, we did not show the data on these metabolites. However, to be thorough, we added it to the supplement of the article.

c) Why didn’t the authors connect the serotonin to specific microbes and their respective numbers?

Response 2c: We have added into the manuscript.

Bacteria affect the secretion of serotonin through influencing enterochromaffin cells. This effect was observed mainly in spore-forming bacteria, such as Clostridium, in the gut microbiome. Moreover, Candida albicans was previously described as a producer of serotonin. However, no correlation was observed between the bacteria and fungi abundance and level of serotonin.

3) Why did the authors use i-FABP and not classical integrity markers like zonulin, alpha 1-antitrypsin or even calprotectin?

Response 3: We intended to measure the gut integrity from the serum of patient. The collected stool was used for metabolome and microbiome analysis. The levels of calprotectin and alpha 1-antitrypsin are measured usually in feces. Therefore, we decided to measure the levels of I-FABP and zonulin. The kit measuring I-FABP is widely used in our laboratory with reliable results. We also measured the levels of zonulin, as was suggested by the reviewer. We repeatedly used the kit from Immundiagnostik and we obtained dubious results. Recently, there are a couple of manuscripts showing that kits for measuring of serum zonulin are not detecting zonulin and they don´t recommend to use it as a marker of mucosal barrier integrity. Therefore, we did not use results from these measurements for this paper.

“Serum zonulin as a marker of intestinal mucosal barrier function:May not be what it seems”. Ajamian, M., Steer, D., Rosella, G., Gibson, PR. PLOS ONE, 2019.

https://doi.org/10.1371/journal.pone.0210728

Widely used commercial ELISA does not detect precursor of haptoglobin2, but recognizes properdin as a potential second member of the zonulin family”. Scheffler, L., Crane, A., Heyne, H. et al. Frontiers in endocrinology. 2018.

https://doi.org/10.3389/fendo.2018.00022

However, here we report our results, which can be potentially added to the manuscript.

4) The authors write several times within the manuscript, that Akkermansia muciniphila produces butyric acid. This is not true. Please see: https://doi.org/10.1099/ijs.0.02873-0

Response 4: We would like to thank the reviewer for this objection. We agree with the reviewer that A. muciniphila does not produce butyric acid. We apologize for this mistake raised from the incorrect source, which was corrected in the manuscript.

5) The often mentioned F/B ratio is scientifically not sound: Finucane et al. 2014 https://doi.org/10.1371/journal.pone.0084689          

Response 5: We would like to thank to the reviewer for this remark. However, there are many recent publications aimed to clarify the relationship between Firmicutes/Bacteroides ratio and BMI (e.g. Louis et al. 2016; Koliada et al. 2017; Zacarias et al. 2018). The authors are convinced that it is quite important to mention F/B ratio since it is obviously affected by the FMT.  

Koliada, A., G. Syzenko, V. Moseiko, L. Budovska, K. Puchkov, V. Perederiy, Y. Gavalko, A. Dorofeyev, M. Romanenko, S. Tkach, L. Sineok, O. Lushchak, and A. Vaiserman. 2017. 'Association between body mass index and Firmicutes/Bacteroidetes ratio in an adult Ukrainian population', BMC Microbiol, 17: 120.

Louis, S., R. M. Tappu, A. Damms-Machado, D. H. Huson, and S. C. Bischoff. 2016. 'Characterization of the Gut Microbial Community of Obese Patients Following a Weight-Loss Intervention Using Whole Metagenome Shotgun Sequencing', PLoS One, 11: e0149564.

Zacarias, M. F., M. C. Collado, C. Gomez-Gallego, H. Flinck, J. Aittoniemi, E. Isolauri, and S. Salminen. 2018. 'Pregestational overweight and obesity are associated with differences in gut microbiota composition and systemic inflammation in the third trimester', PLoS One, 13: e0200305.

6) For many statements within the text, the respective literature is missing (eg. Page 2, l 47; p 2, l 58; p 2, l 59; p2, l 65)

Response 6: All required references were added to the text:

Mendez-Figueroa, V.; Biscaia, J.M.; Mohedano, R.B.; Blanco-Fernandez, A.; Bailen, M.; Bressa, C.; Larrosa, M.; Gonzalez-Soltero, R. Can Gut Microbiota and Lifestyle Help Us in the Handling of Anorexia Nervosa Patients? Microorganisms 2019, 7, doi:10.3390/microorganisms7020058.

Weiss, G. A., and T. Hennet. 2017. 'Mechanisms and consequences of intestinal dysbiosis', Cell Mol Life Sci, 74: 2959-77.

Seitz, J.; Belheouane, M.; Schulz, N.; Dempfle, A.; Baines, J.F.; Herpertz-Dahlmann, B. The Impact of Starvation on the Microbiome and Gut-Brain Interaction in Anorexia Nervosa. Front Endocrinol (Lausanne) 2019, 10, 41, doi:10.3389/fendo.2019.00041.

Gershon, M.D.; Tack, J. The serotonin signaling system: from basic understanding to drug development for functional GI disorders. Gastroenterology 2007, 132, 397-414, doi:10.1053/j.gastro.2006.11.002.

7) In figure 2 the points show a standard deviation. However, the authors state in the Supplement section, that they performed a double measurement only. Hence, a standard deviation is not possible, for this purpose one needs at least three measurements.

Response 7: We thank to the reviewer for this comment. The procedure was not sufficiently described in the Methods. Our statistical analysis was performed based on three independent experiments, i.e. DNA was isolated three times from each stool sample from different parts. Subsequent qPCR were performed in duplicates. Thus, all analysis were calculated from 6 values.

The information was added to the supplementary data.

8) Figure 2, B: Within the text the authors state, that there was no significant difference (p 5, l 208: slight increase). However, in the figure there is an asterix, an indicator for significance? Furthermore, an increase by 3 log shall be significant. Same applies for figure 2, C:

Response 8: We intended to say, that there was only a slight but significant increase of A.m. at the last interval (12m after FMT). Also, our sentence “Similarly, the methanogenic archaeon M. smithii 16S rRNA copies was not altered except the last sample 12 months post-FMT” should give the same informantion, that only last interval was significantly higher.

We corrected these sentences accordingly.

9) Part 3.4 on page 6, line 220 ff: The phylum is Bacteroidetes not Bacteroides!

Response 9: We would like to thank to reviewer, the mistake was corrected in the text.

10) The % in brackets represents the variations during the measurement of 6 respectively 12 months. Please indicate so.

Response 10: The reported variations of % bacterial abundance represent the lowest and highest values obtained during the whole year. The values were not always linear and in some cases the last intervals didn´t represent the lowest/highest values.

The information was added to the text.

11) Page 2, line 227: The shift was seen in Bacteroidetes or Bacteroides?

Response 11: The shift was seen in phylum Bacteroidetes. It was corrected in the text.

12) Page 2, line 247: “insufficient probiotic bacteria” what are these? By definition a probiotic organism is taken by oral route. It is not a taxonomic group!

Response 12: We would like thank to the reviewer for this comment, she/he is definitely right. Our sentence is unsuitable. We have changed the sentence accordingly and added the reference.

Ghouri, Y.A.; Richards, D.M.; Rahimi, E.F.; Krill, J.T.; Jelinek, K.A.; DuPont, A.W. Systematic review of randomized controlled trials of probiotics, prebiotics, and synbiotics in inflammatory bowel disease. Clin Exp Gastroenterol 2014, 7, 473-487, doi:10.2147/CEG.S27530.

13) Page 9, lines 301 ff: The authors did not measure the metabolome but merely SCFA concentrations.

Response 13: The reviewer is correct, we only focused on SCFA, serotonin, tryptophan, kynurenine, and 5-hydroxytryptophan concentrations since these are closely related to microbiome and thus were assumed to be immediately influenced by fecal transplantation. Other metabolites, which can be usually captured by both NMR and MS methods, can be additionally affected by several other non-microbial factors. Therefore, they were not of our interest in this case study.

14) Page 9, Figure 6: What is a normalized intensity. SCFAs are measured in mM

Response 14: Normalized intensity is commonly used in metabolomic studies to express the relative quantity of the given metabolite. For the comparison of metabolites changes between the particular samples within one study, the normalized intensity is sufficient.

We guess that measurement of the absolute concentration would not bring additional information about differences in metabolite quantity between particular samples.

15) Page 10, Figure 7: Why do the authors compare Clostridiales with the SCFA concentration? SCFAs such as Acetate or Propionate are produced by members of all phyla, however butyrate is only produced by members of the Firmicutes. But NOT by members of the Verrucomicrobia as stated in the text!

Response 15: During our analysis, we tried to compare SCFA concentrations with various bacterial orders, however, only Clostridiales positively correlated with SCFA levels. Therefore, we show this correlation. The correlation between SCFA levels and abundance of selected bacterial species from Clostridiales order may suggest a relevant role of these bacteria in the production of SCFA. Nevertheless, other bacteria species also contribute to the pool of SCFA production.

We agree with the reviewer that butyrate is produced only by members of the Firmicutes.

The sentence about the SCFA and Clostridiales was modified.

The statements about A. muciniphila as butyrate-producer were corrected.

Reviewer 2 Report

In this manuscript, Prochazkova et al. examined the gut microbiota, total short-chain fatty acids, serotonin level and gut barrier function in a patient after fecal microbiota transplantation.

The gender of the donor was not clearly specified in the method section. Figure 1: The variation of I-FABP in the sixty ‘healthy’ volunteers seems large. How do the authors define ‘healthy’ here? Do these healthy volunteers also suffer from ‘leaky gut’? What’s the normal range for I-FABP? Figure 6: I would appreciate if the authors provide figures for each short-chain fatty acid, acetate, butyrate and propionate. Fecal metabolome consists of various microbial metabolites. However, the authors only assessed SCFAs and serotonin. At the end of the abstract, the authors mentioned that there were no signs of clinical improvements. But at the end of the introduction, the authors mentioned the clinical gastrological outcome will be reported in another study. This paper just reported some changes in the gut microbiota, barrier function, SCFAs and serotonin after the fecal microbiota transplantation. However, if the readers are curious about how the transplantation affect the clinical outcomes, they need to read another paper. Actually, some readers would like to know whether the fecal transplantation could improve the disease outcome or not, then ask what’s changed after the therapy if they are interested.

Author Response

Response to Reviewer 2 Comments

1) The gender of the donor was not clearly specified in the method section.

Response 1:The stool donor was the patient mother. The information about the gender was added.

2) Figure 1: The variation of I-FABP in the sixty ‘healthy’ volunteers seems large. How do the authors define ‘healthy’ here? Do these healthy volunteers also suffer from ‘leaky gut’? What’s the normal range for I-FABP?

Response 2: The measurement of I-FABP is not a standard clinical procedure, therefore, there is not a set point range determined for the healthy people. Since I-FABP is a marker for intestinal mucosal damage, it is connected to many intestinal diseases. However, it is not a marker for “leaky gut”, but of the mucosal cells damages. Unfortunately, it doesn’t reflect the gut permeability. Very high levels were detected in children with Necrotic enterocolitis (mean 1-20 ng/ml in plasma).

Gut-associated biomarkers L-FABP, I-FABP, and TFF3 and LIT Score for diagnosis of surgical necrotizing enterocolitis in preterm infants. Ng, E., Poon, T., Lam, H. et al. Annals of Surgery. 258(6):1111–1118, DECEMBER 2013. DOI: 10.1097/SLA.0b013e318288ea96

The levels of I-FABP in healthy humans differ a lot in the literature based on the used kit or type of samples. Usually is greatly lower than 1 ng/ml, often undetectable. In our measurement we used sera of healthy women, all screened for chronic inflammation marks and we plotted their values as a box with min to max whiskers and the median. We determined the cutoff based on FDR values<0.5.

The figure was adjusted accordingly.

3) Figure 6: I would appreciate if the authors provide figures for each short-chain fatty acid, acetate, butyrate and propionate. Fecal metabolome consists of various microbial metabolites. However, the authors only assessed SCFAs and serotonin.

Response 3: During our experiments we measured levels of acetate, butyrate and propionate. Since the trends were very similar in all measured SCFA, we decided to show the levels of all SCFA together. Short-chain fatty acids were depicted individually in the supplement. Even though fecal metabolome is composed of various others microbial metabolites, we decided to only focus on acetate, propionate, butyrate, and metabolites of tryptophan pathway, as these are the most frequently connected to microbiota and can be reliably analyzed using NMR and MS platforms.

By MS, we assessed the levels of Serotonin, its precursor 5-Hydroxytryptophan, Tryptophan, and Kinurenin. Because the patient also suffer from depression and evinced specific and non-specific psychiatric symptoms, we believed that FMT could influence the patient psychological clinical status. Therefore we aimed to follow levels of selective neurotransmitters and their precursors, which could be regulated by indigenous bacteria from the gut microbiota.

All additional data were added to the Supplement.

4) At the end of the abstract, the authors mentioned that there were no signs of clinical improvements. But at the end of the introduction, the authors mentioned the clinical gastrological outcome will be reported in another study. This paper just reported some changes in the gut microbiota, barrier function, SCFAs and serotonin after the fecal microbiota transplantation. However, if the readers are curious about how the transplantation affect the clinical outcomes, they need to read another paper. Actually, some readers would like to know whether the fecal transplantation could improve the disease outcome or not, then ask what’s changed after the therapy if they are interested.

Response 4: Our patient with SEAN and SIBO exerted the improvement of the gut microbial dysbiosis after FMT at both bacterial and fungal level. However, her clinical status (gastrointestinal pain and complaints despite all kinds of diets recommended by nutritionist and self-implicated restrictions and other diagnosis specific and non-specific psychiatric symptoms) remained unchanged during all observation period at out-patient visits. Her psychometric properties were evaluated with several Questionnaires. Clinical observations were supported by EDE-Q, BDI II, BAI results from initial and all follow-up periods.

These conclusions were added to the text.

Round 2

Reviewer 2 Report

All comments were addressed.